# Maqui and Chronic Kidney Disease: A Narrative Review on the Potential Nephroprotective Role of Anthocyanins

**DOI:** 10.3390/nu17061058

**Published:** 2025-03-18

**Authors:** Caterina Tiscornia, Violeta Tapia, Daniela Águila, Enrique Lorca-Ponce, Valeria Aicardi, Fabián Vásquez

**Affiliations:** 1Escuela de Nutrición y Dietética, Universidad Finis Terrae, Santiago 7501014, Chile; ctiscornia@uft.cl (C.T.); vtapiap2@uft.edu (V.T.); daguilav@uft.edu (D.Á.); 2Escuela de Enfermería, Universidad Finis Terrae, Santiago 7501014, Chile; elorca@uft.cl; 3Escuela de Kinesiología, Facultad de Arte y Educación Física, Universidad Metropolitana en Ciencias de la Santiago, Santiago 7760197, Chile; 4Unidad de Diálisis, Clínica Indisa, Santiago 7501014, Chile; valeria.aicardi@gmail.com

**Keywords:** antioxidants, anthocyanins, *Aristotelia chilensis*, chronic kidney disease, nephroprotection

## Abstract

**Background/Objectives:** Chronic kidney disease (CKD) is a progressive pathology, with high global prevalence, associated with inflammation and oxidative stress. Given the limited capacity of conventional treatments to reverse renal damage, complementary alternatives have emerged such as supplementation with anthocyanins from maqui (*Aristotelia chilensis*), known for their antioxidant and anti-inflammatory properties. This review analyzes the evidence for their impact on CKD progression. **Methods:** A narrative review of the experimental literature regarding maqui anthocyanins, their bioavailability, and their effects on oxidative stress, inflammation, and CKD to January 2025 was conducted. Articles without peer review or without a focus on *Aristotelia chilensis* were excluded, guaranteeing an updated compilation on its nephroprotective potential. **Results**: Anthocyanins have shown benefits in reducing oxidative stress, inflammation, and glycemia regulation. Preclinical studies suggest improvements in renal function as well as less fibrosis. Human trials indicate positive effects on metabolism, although evidence in CKD patients is limited. Bioavailability remains a challenge to optimizing efficacy. **Conclusions**: Maqui is a promising source of anthocyanins, with nephroprotective potential. However, robust clinical studies are required to determine its safety, optimal dose, and long-term impact in CKD. Its incorporation into evidence-based therapeutic strategies could offer an innovative approach in the management of this disease. More clinical studies are needed to validate the preclinical findings and optimize the therapeutic use of maqui in CKD.

## 1. Introduction

CKD is a pathology of great worldwide relevance, since it is estimated that about 10% of the global population suffers some degree of renal impairment [1]. It is progressive and characterized by the gradual loss of renal function, and it is classified into five stages according to the degree of reduction in glomerular filtration rate (GFR) [2,3,4,5,6]. Among its main causes, diabetes mellitus (DM) is a determining factor, since chronic hyperglycemia damages renal blood vessels, favoring the development of diabetic nephropathy (DN), one of the most frequent forms of renal failure [2,4,7]. On the other hand, arterial hypertension (HT) overloads the glomeruli, accelerating structural deterioration and reducing filtration capacity [5,8,9]. As the disease progresses, the reduction in GFR worsens, increasing the risk of cardiovascular complications and the need for renal replacement therapies (RRTs) such as hemodialysis (HD), peritoneal dialysis (PD), or renal transplantation [1,2].

Likewise, cardiovascular disease (CVD) affects renal perfusion, favoring its progression. Genetic predisposition also influences its development, especially in people with a history of DM or HT, which reinforces the importance of early diagnosis and control of risk factors [7].

Chronic inflammation and oxidative stress play a central role in the pathogenesis and progression of CKD. Persistent renal damage activates diverse immune cell populations, such as macrophages and lymphocytes, which induces the release of proinflammatory cytokines, including interleukin-6 (IL-6), tumor necrosis factor-alpha (TNF-α), and interleukin-1 beta (IL-1β). These molecules perpetuate a sustained inflammatory state, favoring endothelial dysfunction, renal fibrosis, and cell apoptosis [10].

Oxidative stress arises when the balance between reactive oxygen species (ROS) production and endogenous antioxidant capacity is disturbed [11]. Due to its high metabolism and intense mitochondrial activity, the kidney is particularly susceptible to oxidative damage [12,13]. The overproduction of ROS, generated mainly in the mitochondria and through the activity of the enzyme nicotinamide adenine dinucleotide phosphate (NADPH), induces damage to proteins, lipids, and DNA, which aggravates cell injury and accelerates renal dysfunction [14,15,16,17,18,19,20,21]. In advanced stages of CKD, increased oxidative stress is associated with systemic complications such as HT, atherosclerosis, chronic inflammation, and anemia [22,23,24].

A key process in the progression of CKD is renal fibrosis, characterized by myofibroblast activation and an excessive accumulation of extracellular matrix in response to persistent inflammatory and oxidative stimuli. This fibrosis, in addition to compromising renal function, is closely related to vascular deterioration and the progression of DN [25].

In addition, it is important to point out that DM, the main cause of CKD, aggravates these pathological mechanisms. Persistent hyperglycemia generates a sustained inflammatory state and an increase in ROS production, accelerating endothelial damage and renal function deterioration. In addition, the accumulation of uremic toxins, such as indoxyl sulfate (IS) and asymmetric dimethylarginine (ADMA), alters vascular homeostasis by inhibiting the production of nitric oxide (NO), a key regulator of endothelial function [26].

Endothelial dysfunction in CKD promotes atherosclerosis and vascular calcification, processes mediated by phenotypic transformation of vascular smooth muscle cells into osteoblast-like cells, which is exacerbated by hyperphosphatemia and an altered regulation of the extracellular matrix. These alterations not only accelerate the progression of DN but also increase the risk of cardiovascular and peripheral complications, such as difficult-to-heal vascular ulcers, increasing the likelihood of amputations and adverse cardiovascular events [26].

Therefore, given the important relationship between oxidative stress and the progression of CKD, therapeutic strategies aimed at mitigating oxidative stress, reducing inflammation, and improving renal function have been implemented, such as xanthine oxidase inhibitors (such as allopurinol and febuxostat), dietary antioxidants (such as vitamins C and E, coenzyme Q10, resveratrol, and curcumin), NADPH oxidase inhibitors (e.g., GKT137831 and APX-115), and Nrf2 activators (such as bardoxolone methyl) [22,27,28,29,30,31,32,33,34,35,36]. In addition, conventional treatments for early-stage CKD (such as blood pressure control, use of angiotensin receptor blockers, and glycemic control, among others). However, they do not always succeed in reversing renal structural damage, which has prompted the search for complementary alternatives [37,38].

The use of supplements of natural bioactive compounds with antioxidant and anti-inflammatory properties has captured the attention of the scientific community. Anticyanins, flavonoids widely distributed in fruits such as maqui (*Aristotelia chilensis*), have shown promising effects in the protection of renal function by reducing oxidative stress and modulating inflammatory processes [39,40].

Anthocyanins, due to their antioxidant and anti-inflammatory properties, can counteract these processes. In vitro and in vivo studies have shown that anthocyanins can inhibit the production of proinflammatory cytokines, reduce ROS generation, and attenuate oxidative damage in renal cells [37,40].

Maqui, which is rich in anthocyanins, particularly delphinidins, is presented as a potential source of these bioactive compounds for renal protection. Preclinical studies have investigated the effects of maqui extract and its purified anthocyanins in animal models of CKD.

These studies have shown promising results, including improvement of renal function, reduction of inflammation and oxidative stress in the kidney, and attenuation of renal fibrosis. However, it is important to note that most of these studies have been performed in animal models, and further research is needed to confirm these findings in humans [38,41,42,43].

Despite promising preclinical evidence, clinical studies on the effect of maqui anthocyanin supplementation in patients with CKD are limited [44,45]. Well-designed clinical trials, with adequate sample size and sufficient duration, are required to evaluate the efficacy and safety of maqui supplementation in the prevention or treatment of CKD in humans. These studies should consider different stages of CKD, as well as the interaction of anthocyanins with other conventional treatments. In addition, it is essential to investigate the bioavailability and metabolism of maqui anthocyanins in humans to determine the optimal doses and the most effective form of administration.

To date, a comprehensive review on the impact of anthocyanin supplementation in CKD is limited. Therefore, the aim of this narrative review is to summarize the available literature on howreporting that consumption of maqui anthocyanin supplementation may influence CKD progression. In addition, we seek to explore some mechanisms of action of these compounds, together with the results of available preclinical and clinical studies. Finally, the possible therapeutic applications of anthocyanins in the context of CKD will be discussed, identifying research gaps and future directions.

## 2. Materials and Methods

A literature search was conducted using PubMed, Scopus, Google Scholar, MedLine, ScienceDirect, university library’s with the following terms: “Anthocyanins” AND “Maqui” OR “*Aristotelia chilensis*”, “Antioxidants” OR “Oxidative Stress” OR “Inflammation” AND “Chronic Kidney Disease”, “Bioaccessibility” AND “Bioavailability” OR “Delphinidins”, and “Nephroprotective Effects” OR “Renal Fibrosis” AND “Inflammatory Modulation”. Additionally, the reference and “cited by” sections of relevant articles were reviewed to identify further sources. To be included in this narrative review, articles had to meet the following criteria: (1) published before January 2025; (2) reported experimental designs conducted in humans or animal models; (3) focused on the impact of maqui anthocyanin consumption on markers of oxidative stress, inflammation, or renal function; (4) published in English or Spanish; and (5) included studies addressing the bioavailability or bioaccessibility of anthocyanins. Studies not related to *Aristotelia chilensis*, articles not peer-reviewed, and those without a clear focus on chronic kidney disease or oxidative stress were excluded. To ensure a methodological rigor and transparency, a critical appraisal of the included studies was conducted using the Newcastle-Ottawa Scale (NOS) for observational studies and the Cochrane Risk of Bias Tool (RoB) for clinical trials.

This evaluation considered sample size, study design, blinding, confounding factors, and outcome measures to assess study quality and validity. Following the selection of studies, data extraction was performed systematically to ensure consistency and accuracy. The key variables extracted included study design, sample size, intervention type (e.g., maqui extract, purified anthocyanins), dosage, duration, primary outcomes related to oxidative stress, inflammation, and renal function, and any reported adverse effects. Data were synthesized qualitatively, with a focus on identifying the common patterns across studies, evaluating the strength of the evidence, and highlighting discrepancies. Particular attention was given to studies that reported dose–response relationships and the bioavailability of anthocyanins, as these factors are critical for their potential therapeutic application. The findings were then grouped based on study type (preclinical vs. clinical) and relevance to chronic kidney disease, allowing for a comprehensive and balanced narrative synthesis.

Additionally, recent high-impact studies on antioxidants in CKD published within the last three years were prioritized to provide an updated perspective on the field. This approach not only facilitated the inclusion of high-quality evidence but also enhanced the reproducibility and credibility of the findings presented in this review.

The study selection process followed a structured approach. Initially, duplicates were removed, and titles and abstracts were screened for relevance based on predefined inclusion and exclusion criteria. Full-text articles were then assessed for methodological quality, with priority given to randomized controlled trials, systematic reviews, and well-designed observational studies. The discrepancies in study selection or quality assessment were resolved through discussion among the authors to ensure a balanced and unbiased representation of the available evidence. This rigorous methodology enhanced the reliability of the conclusions drawn in this review. To improve accessibility for a multidisciplinary audience, the technical terms were carefully defined within their context, and redundancies were minimized to enhance clarity and coherence. This refinement ensures that the discussion remains scientifically rigorous while being comprehensible to researchers from various fields, including nephrology, nutrition, and biochemistry.

## 3. Complementary Treatments: Antioxidants

### 3.1. Definition of Antioxidants and Anthocyanins

Antioxidants play a fundamental role both in food systems [41] and in the human body to reduce oxidative processes and the harmful effects of ROS [46].

Polyphenols are chemical substances with antioxidant, anti-inflammatory, and prebiotic effects [47]. They are characterized by the presence of one or more phenolic rings and originate through biosynthesis as a secondary product of plant metabolism [47,48].

Within the family of polyphenols, anthocyanins stand out for their participation in vital biological functions due to their antioxidant activity, which is their most important property. They also have antidiabetic, anticancer, anti-inflammatory, antimicrobial, and anti-obesity effects and prevent cardiovascular diseases [46,49,50].

### 3.2. Bioactive Compounds: Anthocyanins in Inflammatory Diseases

Anthocyanins are secondary metabolites and are distributed in flowers, fruits, and vegetables. These components provide various colors such as red, pink, blue, and purple [49]. To date, more than 700 anthocyanins have been identified in nature [51]. There is currently biotechnological research that has focused on the study of plants and their pharmacological properties, examining the functional properties of anthocyanins and how they can interact in humans as anti-inflammatory, antioxidant, anticancer, antidiabetic, and antimicrobial agents, mostly [50,52]. The anti-inflammatory mechanism of anthocyanins is based on their ability to modulate various cellular signaling pathways, including the inhibition of cyclooxygenase-2 (COX-2) expression and the production of proinflammatory cytokines such as TNF-α and IL-6. This modulation of the inflammatory response contributes to protection against chronic diseases associated with inflammation, such as arthritis, cardiovascular diseases, and certain types of cancer [53,54,55,56].

### 3.3. Foods Rich in Anthocyanins: Maqui (Aristotelia chilensis)

*Aristotelia chilensis*, also known as maqui, is a small tree belonging to the Elaeocarpaceae family [53], distributed in the adjacent regions of southern Argentina; it can be found in northern Chile from the province of Limarí to the province of Aysen in southern Chile.

This antioxidant is considered one of the fruits with the highest anthocyanin content [54], which is mainly represented by 8 anthocyanins [57], including delphinidins, powerful antioxidants found in abundance in standardized maqui berry extract [51,53].

Maqui, being rich in anthocyanins, especially delphinidins, offers a high antioxidant potential. Studies have shown that the consumption of maqui or its extracts can contribute to the reduction of oxidative stress, inflammation, and the improvement of metabolic health, which could have positive implications in the prevention of chronic diseases [58,59].

## 4. Properties of Anthocyanins and Their Relevance in CKD

### 4.1. Chemical Composition and Bioactive Properties

Anthocyanins are the main bioactive component of *Aristotelia chilensis* and are also characterized by phytochemical effects, also called secondary metabolites, which give it antioxidant and inflammatory effects [57]. It is worth mentioning that several recent studies have compared the bioactivity of anthocyanins with their catabolites and metabolites from the intestinal microbiota [51]. The evidence has postulated that the bioactivity of anthocyanins is exposed to a variety of catabolites, generating a modulation of inflammatory and cell adhesion pathways [24,37,58].

As noted, CKD is a progressive and irreversible pathology, so it is essential to implement strategies to stop its progression. In this context, supplementation with *Aristotelia chilensis* could be a beneficial option, since it contributes to controlling inflammation and oxidative stress, in addition to exerting a protective effect on the kidney.

Its use in patients with DN corresponding to the initial stages of CKD could help stop the progression of the disease. This would avoid the need for dialysis in advanced stages, reducing the risk of severe kidney failure and improving the patient’s quality of life. Specifically, maqui anthocyanins could protect kidney function by reducing inflammation and oxidative stress, key factors in the progression of CKD. Furthermore, its ability to modulate cell adhesion could have implications in the prevention of renal fibrosis. However, more clinical studies are needed to confirm these effects and establish the optimal doses and long-term safety of maqui supplementation in patients with CKD [60,61,62,63].

### 4.2. Antioxidant and Anti-Inflammatory Mechanism of Action

Recently, Nikbakht et al. (2021) and Li et al. (2015) have shown that the daily administration of 320 mg of anthocyanins for four weeks generates a significant reduction in the parameters of the lipid profile as well as glycemia and insulin resistance, in addition to increasing the levels of HDL cholesterol and serum adipokines. Likewise, it has been identified that anthocyanins have a protective effect against inflammatory processes in chronic pathologies, evidenced by the reduction of proinflammatory markers such as 8-iso-prostaglandins, 3-HODE, and the carbon content in plasma proteins [60,61].

Additionally, its consumption has shown benefits in regulating glycemia in patients with DM and in individuals at risk of developing it, which is especially relevant given that DM is one of the main causes of ND and CKD [60,61]. In particular, cyanidin-3-O-glucoside (C3G) has been studied for its potential impact on renal function in CKD and its relationship with various metabolic pathways [64].

On the other hand, the research by Madduma et al. (2020) analyzed the effect of antioxidant supplementation on oxidative stress and proinflammatory biomarkers in patients with CKD. Although both studies address CKD, they differ in their methodological approaches and in the variables used to evaluate their effects. However, in both cases, foods rich in anthocyanins, such as grapes and blueberries, were administered, and the studies were carried out in animal models with male mice [65].

In the study by Li et al. (2022), developed in an animal model, the effect of C3G on kidney function in CKD was evaluated, observing that, after eight weeks of treatment, there was a reduction in glycemia levels as well as a decrease in the glomerular perimeter and area, in addition to glomerular fibrosis in the groups supplemented with anthocyanins [64]. However, no significant differences were found in the creatinine levels. It is relevant to note that mice with CKD presented greater alterations, and the positive effects were more pronounced in the group that received anthocyanin supplementation [65].

The results obtained by Madduma et al. (2020) were particularly favorable in the group with a high-fat diet supplemented with 5% cranberries, since an improvement in the lipid profile and glycemia, as well as a reduction in the concentration of inflammatory cytokines, was observed. A novel finding of this study was that C3G showed a protective effect on DNA, inhibiting the activation of the transcription factor NF-κB induced by fat-derived palmitic acid. This effect led to a reduction in the expression of proinflammatory cytokines in proximal tubular cells, suggesting that the inhibition of NF-κB generates a protective effect by modulating the immune response and preventing the development of inflammatory pathologies [65].

Along these lines, the study by Qin et al. (2018) complements the evidence on the nephroprotective effect of C3G. Their findings indicate a reduction in blood glucose levels as well as a decrease in the albumin/creatinine ratio (ACR), a reduction in the surface area of Bowman’s capsule and in renal fibrosis, as well as an increase in blood sugar levels of glutathione (GSH), the most potent endogenous antioxidant dependent on selenium [66].

Additionally, Alvarado et al. (2016) and Davinelli et al. (2015) evaluated the effects of anthocyanin supplementation, specifically maqui extract, in overweight, healthy individuals and smokers, analyzing its impact on glucose metabolism in humans with prediabetes. In the study by Alvarado et al. (2016), doses of 60, 80, and 180 mg were administered in 43 participants, while in the study by Davinelli et al. (2015), a double-blind design was used with a single dose of 162 mg/day in the experimental group and a placebo group [67,68].

According to evidence in animal models, anthocyanins, particularly delphinidins and C3G, play a key role in renal protection, regulating altered parameters in CKD, improving ACR and modulating glycemia, suggesting a possible effect of slowing the progression of chronic kidney disease [66,69].

The above is illustrated in Figure 1.

### 4.3. Maqui (Aristotelia chilensis) as a Rich Source of Antioxidants: The Role of Anthocyanins in Its Bioactive Profile

Regarding the content of bioactive compounds in maqui leaves, Crisóstomo et al. (2021) reported the presence of polyphenols and indolic alkaloids in the extracts of this species, including aristoteline, aristotelinine, aristotelone, aristotelinone, aristoquinoline, makonine, and hobartine, in accordance with previous research [70]. In the same study, it was mentioned that other authors identified additional indole alkaloids, such as macomaquina and hobartine, in addition to a quinolic alkaloid, the coumarin scopoletine [70].

The analysis of basal and apical leaves of maqui, both ex vitro in two seasons of the year and in vitro, allowed the identification of 23 bioactive compounds, grouped into five categories of phenolic compounds: derivatives of galloyl acid, caffeoyl-quinic acids, ellagitannins, derivatives of ellagic acid, and flavonoids. The results indicated that, in general, the in vitro leaves of *Aristotelia chilensis* presented a higher concentration of bioactive compounds in four of the five groups mentioned [70].

Likewise, a dichloromethane extract has been identified with a mixture of tri-terpenoids, including ursolic acid and friedelin, in addition to quercetin. It should be noted that the composition of the extracts can vary depending on various environmental factors, such as the collection season, exposure to light, temperature, and type of soil, which influences the diversity and concentration of alkaloids and triterpenoids present in maqui leaves [70].

On the other hand, the maqui fruit is a rich source of delphinidin derivatives, an anthocyanin with high antioxidant power. In addition, it contains other bioactive compounds, such as flavonoids and various phenolic compounds. However, the concentration and composition of these metabolites can vary depending on environmental and agronomic factors, such as growing region, light exposure, temperature, water availability, soil type, seasonality, and the agricultural techniques used. These same factors also influence the variability of the bioactive profile of *Aristotelia chilensis* leaves [57,70].

For years, the content of antioxidant compounds (bioactive) present in the fruit and leaves of the maqui has been investigated. Studies have revealed that the fruit of *Aristotelia chilensis* has a higher concentration of anthocyanins, initially identifying eight main types: delphinidin-3-sambubioside-5-glucoside (Del-3-sa-5-glu), delphinidin-3, 5-diglucoside (Del-3,5-diglu), cyanidine-3-sambubioside-5-glucoside (Ci-3-sa-5-glu), cyanidin-3,5-diglucoside (Ci-3,5-diglu), delphinidin-3-sambubioside (Del-3-sa), delphinidin-3-glucoside (Del- 3-glu), cyanidin-3-sambubioside (Ci-3-sa), and cyanidin-3-glucoside (Ci-3-glu). It was also determined that approximately 34% of the total anthocyanins corresponded to cyanidin derivatives, while the remaining 73% was composed of delphinidin derivatives, the most abundant being Del-3-sa-5-glu [57,71,72].

In addition to anthocyanins, contents of other polyphenols have been described, such as phenolic acids (gentisic acid, ferulic acid, gallic acid, coumaric acid, sinapic acid, 4-hydroxybenzoic acid, and vanillic acid), flavonoids (gallocatechin gallate, quercetin, rutin, myricetin, catechin, epicatechin, and proanthocyanidin B), derivatives of ellagic acid (granatin B, ellagic acid, and rhamnoside ellagic acid), flavonols (myricetin-3-O-galloylglucoside, myricetin-3-O-galactoside, myricetin-3-O-glucoside, quercetin-3-O-rutinoside, quercetin- 3-O-galactoside, quercetin-3-O-glucoside, querceti-na3-O-xyloside, quercetin-3-O-arabinoside, and quercetin-3-Oramnoside), and 5-O-caffeoylquinic acid [57].

Anthocyanins are metabolites that pigment the fruit with its characteristic shiny black color, which delivers its antioxidant power, so if it has a high content of these metabolites, the anti-inflammatory and antioxidant effect is much more effective and powerful [49].

The antioxidant content of a fruit is evaluated using the ORAC (Oxygen Radical Absorption Capacity) method, which quantifies the capacity of a biological sample to neutralize oxygen free radicals [70,71]. By comparing the anthocyanin content of various fresh berries with maqui, both in terms of concentration (mg) and antioxidant capacity per 100 g according to ORAC, it has been determined that maqui is the fruit with the highest anthocyanin content. This conclusion is based on multiple analyses, including HPLC, HPLC-DAD, and the differential pH method. In these evaluations, maqui has been shown to outperform other Chilean berries with high antioxidant capacity, such as calafate and murta, according to the values obtained using ORAC [72,73,74,75,76]. This antioxidant capacity has important implications for health, since it can contribute to protection against oxidative damage and inflammation, processes involved in the development of chronic diseases [77,78,79,80,81,82].

### 4.4. Optimal Dosage and Antioxidant Mechanisms of the Maqui

A recent study carried out a simulated in vitro gastrointestinal digestion in three phases to evaluate the anthocyanins of the maqui fruit. The main finding is the measurement of the percentage of recovery and bioaccessibility in the oral, gastric, and intestinal phases. The recovery percentage allows us to quantify the number of phenolic compounds and polyphenols present in the digest after each phase of food digestion. On the other hand, bioaccessibility refers to the percentage of these compounds that could be available for intestinal absorption after the digestive process [78].

Regarding bioavailability, this concept refers to the fraction of the active substance (in this case, flavonoids and phenols) that reaches the bloodstream and is available to exert its effect after being administered [79].

The results obtained in the study show the following recovery values of total phenols (TPs) and total flavonoids (TFs) in each digestive phase: oral phase, TF 103% and TP 79%; gastric phase, TF 102% and TP 112%; and intestinal phase, TF 17% and TP 21% [78].

For its part, the bioaccessibility in the intestinal phase was TF 78% and TP 14%, while the bioavailability of these compounds, measured in the last digestive phase, also corresponded to TF 78% and TP 14% [78]. These results indicate that the bioactive compounds are released mainly in the oral and gastric phases, as evidenced by the high recovery percentages (103% and 102%, respectively). This phenomenon is attributed to the breaking of bonds of bioactive compounds due to the action of digestive enzymes such as α-amylase in the oral phase and pepsins in the gastric phase [78].

However, in the intestinal phase, the bioaccessibility of these compounds is significantly reduced, observing a decreased stability of anthocyanins, with a recovery percentage of only 17%. This reduction is because, after exposure to gastrointestinal conditions, anthocyanins undergo structural modifications that alter their chemical properties, which decreases their bioavailability and, therefore, their potential bioactive effect [78].

Moreover, Guo, Y. et al. (2020) evaluated the dose–response relationship of supplementation with purified anthocyanins in healthy young adults, analyzing its impact on inflammatory markers, oxidative stress, and metabolic risk factors. A randomized, double-blind, placebo-controlled clinical trial was conducted in 111 participants between 18 and 35 years of age, who received daily doses of 20, 40, 80, 160, or 320 mg of anthocyanins or placebo for 14 days. The results showed that the 80 mg/day dose significantly reduced fasting glucose (*p* = 0.007) and levels of 8-iso-PGF2α, a marker of oxidative stress (*p* = 0.009), while lower doses (≤40 mg/day) were not effective, and higher doses (≥160 mg/day) offered no additional benefit. A decrease in interleukin-10 (IL-10) was observed with increasing doses of anthocyanins (*p* = 0.025), suggesting an anti-inflammatory effect, although no significant changes were found in TNF-α and IL-6. All doses were well tolerated, and no relevant adverse effects were reported. In conclusion, supplementation with 80 mg/day of anthocyanins appears to be the minimum effective dose to improve glucose homeostasis and reduce oxidative stress in healthy adults, supporting its potential preventive use in metabolic diseases [83].

Various studies have investigated the effects of maqui berry extract (*Aristotelia chilensis*) on metabolic and oxidative stress parameters. Davinelli et al. (2015) conducted a randomized, placebo-controlled clinical trial in which 42 participants (ages 45 to 65 years) consumed 162 mg of maqui-derived anthocyanins three times a day for four weeks. The results showed a significant reduction in plasma levels of oxidized low-density lipoproteins (Ox-LDLs) and in urinary levels of F2-isoprostanes (8-iso-PGF2α), markers of oxidative stress [68]. For their part, Alvarado et al. (2016) administered maqui extract capsules with 60, 120, and 180 mg of anthocyanins to prediabetic individuals with a family history of type 2 DM, high blood pressure, body mass index greater than 23 kg/m^2^, or dyslipidemia. The study found that a 60 mg dose decreased fasting glucose, while a 180 mg dose significantly reduced fasting insulin levels, suggesting that higher doses may be more effective in preventing type 2 diabetes [67]. These findings support the therapeutic potential of maqui extract in improving metabolic parameters and reducing oxidative stress in at-risk populations.

A critical evaluation of the included studies was conducted to assess their methodological strengths and limitations. While several preclinical studies demonstrated promising nephroprotective effects of maqui anthocyanins through reductions in oxidative stress, inflammation, and fibrosis, many were conducted in animal models, limiting their direct translatability to human populations. Furthermore, differences in study design, including variations in anthocyanin doses, extraction methods, and duration of interventions, introduce heterogeneity that complicates direct comparisons. Among the human clinical trials, sample sizes were often small, and study durations were short, making it difficult to establish long-term efficacy and safety. Some studies lacked adequate blinding or randomization, increasing the risk of bias. Additionally, bioavailability remains a key challenge, as anthocyanins undergo significant degradation during digestion, potentially affecting their therapeutic potential. Despite these limitations, the consistency of findings across different models supports further investigation through well-designed, long-term clinical trials to validate the nephroprotective effects of maqui anthocyanins in patients with chronic kidney disease.

The following Table 1 summarizes the main studies and their findings.

## 5. Discussion

CKD is a prevalent condition worldwide, characterized by a silent progression that gradually deteriorates renal function, significantly affecting patients’ quality of life. In this context, interest in complementary therapeutic strategies that can slow its progression has increased. This narrative review analyzes the potential of supplementation with anthocyanins derived from maqui (*Aristotelia chilensis*) as a nutritional intervention with possible nephroprotective effects [1,2].

The evidence reviewed suggests that anthocyanins, thanks to their antioxidant and anti-inflammatory properties, could contribute to the reduction of oxidative stress, a key factor in the progression of CKD [60,61,65]. Anthocyanins have demonstrated the ability to modulate proinflammatory pathways and improve redox homeostasis, which could translate into a protective effect on renal function [46,50].

Several preclinical studies have shown the positive effects of anthocyanins in experimental models of CKD, evidencing a reduction in biomarkers of kidney damage, better regulation of blood glucose, and a decrease in systemic inflammation [63,64,65,66,67,68,69].

Compounds such as C3G and delphinidins, present in high concentrations in maqui, have been the subject of research, suggesting their potential to slow renal deterioration [64]. However, although animal models have shown promising results, evidence in humans remains insufficient and requires well-designed clinical studies to validate these effects in CKD patients.

In terms of bioavailability, studies indicate that anthocyanins undergo significant degradation along the gastrointestinal tract, which represents a challenge to optimizing their absorption and therapeutic efficacy. Although formulations of standardized maqui extracts have shown positive metabolic effects in human studies, heterogeneity in methodological designs, doses used, and characteristics of the subjects studied make it difficult to formulate precise clinical recommendations [78,79,83]. In this regard, it is crucial to develop research that determines the optimal dose, long-term safety profile, and interactions with other conventional treatments for CKD.

From an applied perspective, supplementation with maqui anthocyanins could represent an adjuvant therapeutic approach in the management of CKD, especially in patients with metabolic comorbidities such as DM and HT, primary risk factors in disease progression [7,25]. The inclusion of natural antioxidants in the dietary management of CKD could contribute to improving the inflammatory and oxidative status of patients, favoring a comprehensive and personalized approach in the treatment of the disease.

Compared to other international studies, the results obtained in this maqui anthocyanin research are consistent with the findings of other research groups that have evaluated the effects of antioxidants in the context of CKD [11,22,25,58]. It has been shown that these compounds can modulate oxidative stress and decrease inflammation in experimental models, suggesting their potential to slow the progression of kidney disease [68]. However, although the effects in animal models have been promising, human studies remain limited, and further clinical trials are needed to validate their applicability in CKD patients.

Additionally, research by Verma et al. (2021) and other groups in Europe and North America have documented the positive effects of antioxidants such as vitamin C, E, and uric acid on the progression of CKD [22,64,65,66,67]. These antioxidants have shown similar benefits in reducing the markers of oxidative stress and improving renal function, reinforcing the hypothesis that maqui anthocyanins could have similar effects [50,58,60]. However, the bioavailability of these antioxidants remains a challenge, as observed in studies on vitamin C and E, whose absorption and effectiveness are limited by their metabolism in the gastrointestinal tract [64,78,79]. This is an aspect that should also be considered in the development of maqui therapies, which underlines the need to improve the formulation of these compounds.

Despite the growing interest in the use of bioactive compounds such as anthocyanins in nephroprotection, the current evidence is not yet sufficient to recommend their widespread clinical implementation. Controlled clinical trials, with representative samples and long-term follow-up, are essential to evaluate both the efficacy and safety of maqui anthocyanin supplementation in the progression of CKD. These studies, in addition to broadening the understanding of its applicability, will allow for the determination of its role within a comprehensive therapeutic approach for patients, considering individual variability and interactions with other conventional treatments.

## 6. Conclusions

In conclusion, supplementation with maqui anthocyanins emerges as a promising alternative in reducing kidney damage associated with CKD, thanks to its antioxidant and anti-inflammatory capacity. However, its clinical application requires greater scientific substantiation based on studies in humans that confirm its impact on the evolution of the disease. In the future, research in this field should focus on the elucidation of their specific mechanisms of action, the determination of effective and safe doses, and the integration of these compounds into evidence-based therapeutic strategies for the management of CKD.

## Figures and Tables

**Figure 1 nutrients-17-01058-f001:**
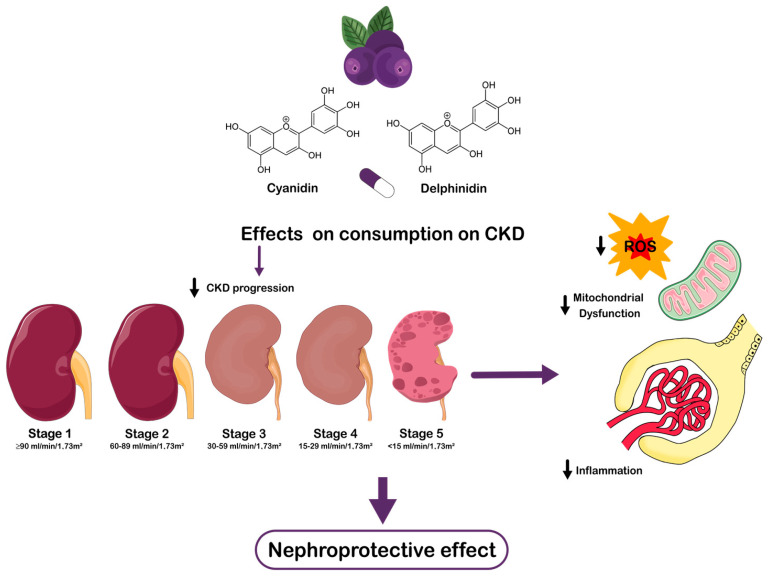
Antioxidant and anti-inflammatory effects in CKD.

**Table 1 nutrients-17-01058-t001:** Principal studies on the use of polyphenols and their protective role in CKD and other diseases.

Authors	Type of Source	Aim	Principal Findings
Kwon, et al., 2017 [34]	Research Article	To evaluate APX-115’s efficacy in mitigating diabetic kidney injury.	APX-115 prevented kidney injury, oxidative stress, and organelle dysfunction in diabetic mice, comparable to losartan, suggesting its potential as a therapeutic agent for diabetic kidney disease.
Cha, et al., 2017 [35]	Research Article	To evaluate the therapeutic efficacy of APX-115 in diabetic nephropathy.	APX-115 reduced oxidative stress, improved renal function, and attenuated mesangial expansion in diabetic mice. It demonstrated superior or comparable efficacy to GKT137831, suggesting pan-Nox inhibition as a potential treatment for diabetic nephropathy.
Céspedes, et al., 2008 [59]	Research Article	To evaluate antioxidant and cardioprotective effects of *Aristotelia chilensis* fruit extract.	The methanol extract of *Aristotelia chilensis* demonstrated significant antioxidant activity, cardioprotective effects against ischemia/reperfusion injury, and reduced lipid peroxidation, correlating with high polyphenol content.
Nikbakht, et al., 2021 [60]	Clinical Trial	To evaluate the anti-inflammatory effects of dietary anthocyanin in type 2 diabetic, at-risk, and healthy individuals.	Dietary anthocyanin significantly reduced pro-inflammatory biomarkers in type 2 diabetic participants and improved select biochemical parameters in at-risk individuals.
Li, et al., 2015 [61]	Randomized Controlled Trial	To evaluate anthocyanins’ effects on dyslipidaemia, oxidative status, and insulin sensitivity in type 2 diabetes patients.	Anthocyanin supplementation improved lipid profiles, enhanced antioxidant capacity, reduced oxidative stress markers, and improved insulin sensitivity and glucose metabolism in type 2 diabetes patients.
Li, et al., 2022 [64]	Research Article	To investigate anthocyanins’ effects on diabetic kidney disease via metabolic pathways.	Anthocyanins significantly improved renal function, reduced blood glucose, and alleviated glomerular lesions in DKD mice by regulating amino acid metabolism, particularly taurine, hypotaurine, tryptophan, and tyrosine pathways.
Qin, et al., 2018 [66]	Research Article	To investigate the effects of cyanidin 3-glucoside on diabetic nephropathy in db/db mice.	Cyanidin 3-glucoside ameliorates diabetic nephropathy by reducing glucose metabolic dysfunction, renal inflammation, fibrosis, and oxidative stress, while enhancing glutathione synthesis in db/db mice.
Alvarado, et al., 2016 [67]	Clinical Trial	To evaluate Delphinol®’s effects on glucose metabolism and lipid profiles in prediabetic subjects.	Delphinol® significantly reduced HbA1c and LDL levels, increased HDL, and improved glucose metabolism over three months, with no adverse effects observed. Fasting insulin and glucose changes were non-significant.
Davinelli, et al., 2015 [68]	Randomized Controlled Trial	To evaluate maqui berry extract’s impact on lipid peroxidation biomarkers	Delphinol® supplementation reduced Ox-LDL and urinary F2-isoprostanes at 4 weeks, but effects diminished by 40 days. No significant changes in anthropometrics, blood pressure, or lipid profile were observed.
Crisóstomo-Ayala, et al., 2021 [70]	Research Article	To compare bioactive compounds in maqui leaves from in vitro and ex vitro sources across developmental stages and seasons.	In vitro maqui leaves exhibited higher total phenolic content, while winter basal leaves had higher flavonoid content. Spring basal leaves were enriched in quercetin, catechin, kaempferol, and 3-caffeoyl quinic acids, whereas in vitro leaves contained α-tocopherol and β-sitosterol. Adult leaves showed elevated linolenic and linoleic acids, indicating potential antioxidant and nutraceutical applications.
Lucas-González, et al., 2016 [78]	Research Article	To evaluate the impact of in vitro gastrointestinal digestion on maqui berry polyphenolic stability and antioxidant activity.	Gastrointestinal digestion significantly reduced polyphenolic concentrations, particularly anthocyanins, and decreased antioxidant scavenging properties. However, chelating activity increased, and phenolic and flavonoid bioaccessibility remained at 78.19% and 14.20%, respectively, indicating retained antioxidant potential.
Guo, et al., 2020 [83]	Randomized Controlled Trial	To evaluate the dose-response relationship of anthocyanins on metabolic and inflammatory biomarkers.	Anthocyanin supplementation (>80 mg/d) significantly reduced fasting plasma glucose, increased interleukin-10 levels, and decreased 8-iso-prostaglandin F2α, demonstrating antioxidant and anti-inflammatory effects in healthy adults.

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
