# Peer review of "Maqui and Chronic Kidney Disease: A Narrative Review on the Potential Nephroprotective Role of Anthocyanins"

_nutrients, 2025, doi:10.3390/nu17061058_

Round 1
Reviewer 1 Report
Comments and Suggestions for Authors
Article Title: Maqui and Chronic Kidney Disease: A Narrative Review on the Potential Nephroprotective Role of Anthocyanins
Summary of the Article: The manuscript presents a narrative review on the impact of maqui anthocyanin supplementation on the progression of chronic kidney disease (CKD). It covers the pathophysiological aspects of CKD, the role of oxidative stress and inflammation in its progression, and the therapeutic potential of maqui bioactive compounds. Preclinical and clinical studies are reviewed, highlighting the need for further research to validate its efficacy in humans.
Content Evaluation
1. Originality and Relevance The study addresses a topic of increasing interest in the scientific and clinical community, given the high prevalence of CKD and the need for effective complementary therapies. The review on maqui anthocyanins and their potential impact on CKD is novel and provides valuable information for the field of nephrology and nutrition.
2. Structure and Coherence The manuscript is well-structured, with a logical organization that facilitates reader comprehension. The introduction provides adequate context on CKD, justifying the need to explore alternative therapies. However, some sections present redundancies, especially in the discussion of oxidative stress and inflammation.
3. Quality of Literature Review The manuscript compiles a broad range of scientific literature, including relevant preclinical and clinical studies. However, a more critical evaluation of the methodologies employed in the cited studies would strengthen the validity of the claims made. Additionally, some key recent studies on antioxidants in CKD could be incorporated.
4. Methodology of Literature Search The materials and methods section adequately describes the sources consulted and the study selection criteria. However, there is no information on the selection and quality assessment process of the included studies, which could affect the reproducibility of the review.
5. Clarity and Precision of Scientific Language The language is appropriate for an academic publication, although some sections contain repetitions and excessive use of technical terms without contextual explanation. It is recommended to improve the writing to make the text more accessible to a multidisciplinary audience.
6. Conclusions and Recommendations The conclusions adequately summarize the findings presented in the review and highlight the need for more clinical studies. However, the discussion on clinical applicability and current limitations in translating preclinical findings to human settings could be reinforced.
Recommendations for Improvement
- Avoid redundancies in the discussion on oxidative stress and inflammation.
- Include a critical analysis of the cited studies, evaluating their strengths and methodological limitations.
- Expand the discussion on bioaccessibility and bioavailability of anthocyanins in humans, considering possible strategies to improve their absorption.
- Clarify the study selection methodology, detailing inclusion and exclusion criteria, as well as the quality assessment of the studies.
- Revise the writing to enhance clarity and precision of scientific language, avoiding unnecessary repetitions.
- Further explore the clinical applicability of the findings and the potential mechanisms of action in humans.
Final Decision
Recommendation: Revise and Resubmit after incorporating the suggested improvements.
Justification: The manuscript addresses a relevant topic and has strong bibliographic support. However, it requires a more critical analysis of the included studies, as well as improvements in structure and language clarity. A thorough revision could strengthen its academic impact and its potential for publication in a high-impact journal.
Author Response
We sincerely appreciate the reviewers' feedback on our manuscript. We have carefully reviewed all the comments and made the necessary modifications to the original manuscript to address the suggestions provided.

Reviewer 2 Report
Comments and Suggestions for Authors
Dear Authors,
I am pleased to review your manuscript titled: “Maqui and Chronic Kidney Disease: A Narrative Review on the Potential Nephroprotective Role of Anthocyanins.” The manuscript addresses a highly relevant topic. Below are my comments:
INTRODUCTION:
The Introduction section is well-written. I suggest expanding the paragraph on Chronic inflammation and oxidative stress, providing a more detailed explanation of the underlying mechanisms. Additionally, it would be beneficial to clarify the link between these factors and diabetes, as this condition is the leading cause of chronic kidney disease (CKD). The vascular damage associated with chronic inflammation and oxidative stress is often linked to diabetes, leading not only to kidney damage but also to complications such as vascular issues, with important implications like vascular ulcers. You may find supporting references for this in the following article: https://doi.org/10.3390/diabetology6020010
https://doi.org/10.1007/s00109-021-02037-7
METHODS:
A narrative review does not require a structured methodological approach. However, I would ask you to include, at the end of the Methods section, a description of how the data were extracted and synthesized.
RESULTS:
The aim of your review is: “to summarize the available literature on how consumption of maqui anthocyanin supplementation may influence CKD progression. In addition, we seek to explore some mechanisms of action of these compounds, together with the results of available preclinical and clinical studies. Finally, possible therapeutic applications of anthocyanins in the context of CKD will be discussed, identifying research gaps and future directions.”
Based on this objective, the results section should exclusively address this aim. Therefore, Sections 3.1 to 3.5 should be removed. Some information can be included in the Introduction, but only the essential points. For example, the classification of CKD is unnecessary. I recommend focusing solely on the core objective.
DISCUSSION:
This section should follow the Results section. The authors should discuss their findings in comparison with other international studies to strengthen their conclusions. Additionally, a section on Implications for clinical practice should be included.
CONCLUSIONS:
The conclusions are quite lengthy. This section should be shortened, and part of the text could be incorporated into the Discussion section.
Author Response

(The authors gave the same response as above.)

Round 2
Reviewer 1 Report
Comments and Suggestions for Authors
The authors have responded adequately to all requests and comments. The article could be published in this format.
Reviewer 2 Report
Comments and Suggestions for Authors
The manuscript has been extensively edited by the authors with appropriate changes to suggestions. I believe it can be accepted for publication